# Enhancing Postharvest Quality and Antioxidant Capacity of Blue Honeysuckle cv. ‘Lanjingling’ with Chitosan and Aloe vera Gel Edible Coatings during Storage

**DOI:** 10.3390/foods13040630

**Published:** 2024-02-19

**Authors:** Jinli Qiao, Dalong Li, Liangchuan Guo, Xiaoqi Hong, Shuman He, Junwei Huo, Xiaonan Sui, Yan Zhang

**Affiliations:** 1College of Horticulture and Landscape Architecture, Northeast Agricultural University, Harbin 150030, China; 2Key Laboratory of Biology and Genetic Improvement of Horticultural Crops (Northeast Region), Ministry of Agriculture and Rural Affairs, Northeast Agricultural University, Harbin 150030, China; 3National-Local Joint Engineering Research Center for Development and Utilization of Small Fruits in Cold Regions, Northeast Agricultural University, Harbin 150030, China; 4Heilongjiang Green Food Science Research Institute, Northeast Agricultural University, Harbin 150030, China; 5College of Food Science, Northeast Agricultural University, Harbin 150030, China

**Keywords:** Haskap, edible coating, aloe vera gel, postharvest storage, antioxidant capacity

## Abstract

This study investigated the impact of chitosan (CH, 1%) and aloe vera gel (AL, 30%) edible coatings on the preservation of blue honeysuckle quality during a 28-day storage at −1 °C. Coating with CH, AL, and CH+AL led to notable enhancements in several key attributes. These included increased firmness, total soluble solids, acidity, pH, and antioxidant capacity (measured through DPPH, ABTS, and FRAP assays), as well as the preservation of primary (ascorbic acid) and secondary metabolites (TPC, TAC, and TFC). The TAC and TFC levels were approximately increased by 280% and 17%, respectively, in coated blue honeysuckle after 28 d compared to uncoated blue honeysuckle. These coatings also resulted in reduced weight loss, respiration rate, color, abscisic acid, ethylene production, and malondialdehyde content. Notably, the CH+AL treatment excelled in preserving secondary metabolites and elevating FRAP-reducing power, demonstrating a remarkable 1.43-fold increase compared to the control after 28 days. Overall, CH+AL exhibited superior effects compared to CH or AL treatment alone, offering a promising strategy for extending the shelf life and preserving the quality of blue honeysuckle during storage.

## 1. Introduction

Blue honeysuckle (*Lonicera caerulea* L.) fruit is hailed as a “homology of medicine and food” due to its rich reservoir of bioactive compounds and nutritional properties [1]. It has demonstrated a diverse range of biological activities, including anti-lung injury, anti-cancer, anti-obesity, anti-viral, anti-bacterial, anti-diabetic, and radiation resistance [2]. In recent years, blue honeysuckle has gained immense popularity, particularly among consumers in China, Poland, and Russia. By 2023, China’s blue honeysuckle cultivation area expanded to 80,000 mu, yielding a maximum of 40,000 tons from 5-year-old fruit trees. In Canada, Russia, and Japan, the cultivation areas stand at 15,000, 11,025, and 2400 mu, respectively [3,4]. Nevertheless, blue honeysuckle has a brief harvesting window due to its pronounced seasonality, primarily in June and July. Its thin skin and high moisture content make postharvest preservation challenging, leading to over 90% of the produce either being frozen or processed into various products. With the increasing consumer demand for nutritious, fresh foods, there is a pressing need for more effective postharvest preservation methods for blue honeysuckle.

Physical preservation methods, including a controlled atmosphere, irradiation, and edible coatings, are commonly employed. Among these, atmosphere and irradiation preservation can be cost-intensive, while edible coatings offer an eco-friendly, cost-effective, and easy-to-apply alternative [5]. Edible coatings create a thin film layer that regulates moisture and air transport, thus preserving fruit quality and extending shelf life [6]. These coatings typically use biodegradable, biocompatible, and non-toxic materials such as polysaccharides, lipids, and proteins [5]. Chitosan (CH), a natural macromolecule polysaccharide, is an optimal choice for postharvest coatings due to its biocompatibility, biodegradability, and bioactivity [5]. Numerous studies have demonstrated the efficacy of CH in preserving berries. For instance, a 1% CH coating on grapes reduced weight loss and enhanced the activity of antioxidant enzymes like superoxide dismutase and ascorbate peroxidase [7]. Similarly, a 0.5% CH treatment of blueberries reduced weight loss [8]. CH coatings also serve as effective carriers for antimicrobial and antioxidant functions. However, CH is not as proficient as aloe vera gel (*Aloe barbadensis* Miller) in reducing moisture loss, making aloe vera gel preservation an emerging area of interest [9]. Recent studies have highlighted the effectiveness of aloe vera gel (AL) in reducing moisture loss and inhibiting microorganism proliferation in blueberries [8] and table grapes [9]. A 0.5% AL coating, by incorporating lipids into the composite, can enhance hydrophobic properties and barrier effectiveness [8]. Considering the complementary properties of CH and AL, their sequential use in preserving blue honeysuckle might yield superior results. To the best of our knowledge, a combined CH and AL coating for blue honeysuckle preservation during cold storage has not been previously investigated.

Hence, this study aims to explore the effects of CH and AL coatings, with enhanced barrier properties, on the quality attributes of blue honeysuckle cv. ‘Lanjingling’ during 28 days of storage at −1 °C. We examine factors such as firmness, weight loss, respiration rate, color, metabolites, and antioxidant capacity. This research opens new avenues for the development of effective edible coatings for post-harvest fruit preservation and offers valuable insights for the selection of suitable coatings for blue honeysuckle.

## 2. Materials and Method

### 2.1. Materials

Chitosan with a molecular weight of 340 kDa and a degree of deacetylation exceeding 85% was purchased from Yuanye BIO-Technology Co., Ltd. (Shanghai, China). Gallic acid standard (GA), abscisic acid standard (ABA), ferrous sulfate heptahydrate standard (FeSO_4_·7H_2_O), ascorbic acid standard (ASA), and 6-hydroxy-2,5,7,8-tetramethylchromane-2-carboxylic acid (Trolox) were obtained from Yuanye BIO-Technology Co., Ltd. (Shanghai, China). For our high-performance liquid chromatography (HPLC) needs, HPLC-grade potassium dihydrogen phosphate, methanol, and acetonitrile were obtained from Xingmake Science and Technology Development Co., Ltd. (Tianjin, China). Purified water was purchased from Watsons Food and Beverage Co., Ltd. (Guangzhou, China). Hydrochloric acid and glacial acetic acid were acquired from Fuyu Fine Chemical Co., Ltd. (Tianjin, China). Folin–Ciocalteau, 2,2-diphenyl-1-picrylhydrazine (DPPH) were purchased from Bomei Biotechnology Co., Ltd. (Anhui, China). Sodium carbonate, sodium hydroxide, potassium chloride, ferric chloride hexahydrate (FeCl_3_·6H_2_O), and sodium acetate were acquired from Kemio Chemical Reagent Co., Ltd. (Tianjin, China). 2,2′-Azino-bis-(3-ethylbenzothiazoline-6-sulfonic acid) diamine salt (ABTS) and 2,4,6-tri (2-pyridyl)-s-triazine (TPTZ) were acquired from Biotopped Technology Co., Ltd. (Beijing, China). Aloe vera leaves (*Aloe barbadensis* Miller) with a 60–70 cm average height were obtained from a local flower market (Harbin, China). Blue honeysuckle cv. ‘Lanjingling’, resulting from a hybridization between (*L. caerulea* subsp. *altaica* × *L. caerulea* subsp. *kamtschatica*) and *L. caerulea* subsp. *kamtschatica* F_2_, was meticulously harvested at its optimal commercial ripeness (45 d after flowering stage) at Xiangyang base station (39.93° N latitude, 116.04° E longitude) in Harbin, China.

### 2.2. Preparation of Edible Coating Solution

The chitosan (CH) edible coating solutions were prepared following the method outlined by Jiang et al. [10]. In brief, chitosan (20 g) was dissolved in 2 L of acetic acid solution (0.5%, *w*/*w*) to create a 1% chitosan solution. Subsequently, 30% *w*/*w* (based on chitosan) glycerol was added as a plasticizer, along with 5% *w*/*w* (based on chitosan) as a surfactant. The pH of this solution was then adjusted to pH 3 using glacial acetic acid, which was monitored using a pH meter (S400-K, Mettler Toledo Inc., Greifensee, Switzerland).

The aloe vera gel (AL) edible coating solutions were prepared following the procedure outlined by Nia et al. [9]. Firstly, 10 g of sodium hydroxide was added to 500 mL of deionized water to prepare a 2% NaOH solution. The aloe vera leaves were rinsed with the NaOH solution, after which the gel was separated from the outer skin layer of the leaves and blended (SP526, Supor Co. Ltd., Zhejiang, China) into a fine pulp. Subsequently, the gel was pasteurized in a water bath at 65 °C for 30 min and promptly cooled. This pasteurization process was repeated three times, and the resulting solutions were stored at 4 °C. The mixture was then filtered to eliminate any remaining fibers. Finally, the gel was diluted with distilled water (30:70, *v*/*v*), and 1% *v*/*v* glycerol was added to enhance plasticity.

### 2.3. Treatment and Storage of Blue Honeysuckle

Blue honeysuckle cv. ‘Lanjingling’, selected for their uniform maturity and absence of any mechanical damage or fungal decay, were randomly assigned to four distinct groups: the control group received no coating treatment; Treatment 1 involved a 1% CH treatment, where blue honeysuckle was submerged in a chitosan solution for 5 min and subsequently air-dried at 25 °C for 12 h; Treatment 2 utilized a 30% AL treatment, immersing blue honeysuckle in an aloe vera gel solution for 20 s before air-drying the honeysuckle at 25 °C for 12 h; Treatment 3 comprised a dual application, with blue honeysuckle first immersed in a CH solution and then in an AL solution. Blue honeysuckle was divided into 72 boxes (food-grade plastic box with high air permeability and perforations) and stored at −1 °C with approximately 80% relative humidity for a duration of 28 days. Physicochemical parameters and antioxidant capacity were measured at six time points: 0, 3, 7, 14, 21, and 28 days. At each sampling point, three boxes of blue honeysuckle were selected, with each box serving as an individual replicate (500 g per replicate).

### 2.4. Wax Coverage, Weight Loss, and Firmness of Blue Honeysuckle

The epicuticular wax levels of blue honeysuckle were categorized into five grades: 0, indicating no wax coverage; 1, denoting wax coverage ranging from 0 to 1/3; 2, representing wax coverage between 1/3 and 2/3; 3, indicating wax coverage ranging from 2/3 to 1; and finally, 4, signifying complete wax coverage. The calculation of epicuticular wax coverage is determined by Equation (1):(1)$$\mathrm{Wax}\;\mathrm{coverage}\;\left(\%\right)=\frac{\sum \left(\mathrm{fruit}\;\mathrm{quantity}\times \mathrm{fruit}\;\mathrm{wax}\;\mathrm{level}\right)}{4\times \mathrm{total}\;\mathrm{fruit}\;\mathrm{quantity}}\times 100\%$$

The weight loss of blue honeysuckle was assessed on day 0 and subsequently at designated storage intervals (3, 7, 14, 21, and 28 days). It was expressed as a percentage relative to the initial weight and calculated using Equation (2):(2)$$Y\left(\%\right)=\frac{W_{1}-W_{0}}{W_{0}}\times 100\%$$
where *W*_1_ represents the weight at each storage interval; *W*_0_ represents the weight of the initial day.

Firmness was determined using a hardness tester (FHM-5, Takemura Electric Work. Co. Ltd., Tokyo, Japan) equipped with an 8 mm probe, following the instructions provided in the manual. The probe was lowered at a consistent speed until it reached the bottom of the blue honeysuckle to measure its firmness.

### 2.5. Total Soluble Solids (TSS), Acidity, and pH of Blue Honeysuckle

TSS, acidity, and pH were evaluated as fundamental parameters in the postharvest preservation test. The TSS and acidity levels were determined using a portable Brix-acidity meter (PAL-BX/ACID7, ATAGO Co. Ltd., Tokyo, Japan). The pH of blue honeysuckle was measured using a pH meter.

### 2.6. Respiration Rate and Color of Blue Honeysuckle

The respiration rate was measured using a fruit and vegetable respirometer (SYS-GH30A, SAIYASI Technology Co. Ltd., Shenzhen, China). The results were calculated and expressed in milligrams of carbon dioxide produced per kilogram per hour (mg CO_2_/kg • h).

The color values of the blue honeysuckle were determined using a chroma meter (CR-400, Konica Minolta Inc., Osaka, Japan) with an illuminant C and a target mask of 6 mm diameter. Zero correction was performed by measuring a white calibration cap, and ten measurements were performed for each area. The color results were expressed as CIE *L**, *a**, *b**, and the total color difference (Δ*E*) was calculated using Equation (3).
(3)$$\Delta E=\sqrt{{\left({L^{*}}_{sample}-{L^{*}}_{0}\right)}^{2}+{\left({a^{*}}_{sample}-{a^{*}}_{0}\right)}^{2}+{\left({b^{*}}_{sample}-{b^{*}}_{0}\right)}^{2}}$$
where Δ*E* represents chromatic aberration; *L*^*^*_sample_* and *L*^*^_0_ are brightness values; *a*^*^*_sample_* and *a*^*^_0_ represent red and green values; *b*^*^*_sample_* and *b*^*^_0_ represent yellow and blue values.

### 2.7. Determination of Ascorbic Acid (ASA) Content Using High-Performance Liquid Chromatography Diode Array Detector (HPLC-DAD)

The ASA content was determined following the procedure outlined by Klimczak and Gliszczyńska-Świgło [11]. The analysis was conducted using an HPLC system comprising an Agilent 1200 instrument, which was equipped with a diode array detector (Agilent 1200 DAD WR, Agilent Technologies Inc., Palo Alto, CA, USA). Separation was carried out on a C18 column (150 mm × 4.6 mm, 5 μm, Shiseido Co. Ltd., Tokyo, Japan). A gradient solvent profile was employed, consisting of solvent A (5 mmol/L KH_2_PO_4_) and solvent B (methanol): 0 min, 95% A; 6 min, 78% A; and 9 min, 95% A. The column was maintained at 25 °C, with a flow rate of 0.8 mL/min, and the detector wavelength was set at 245 nm.

### 2.8. Determination of Total Polyphenol Content (TPC), Total Flavonoid Content (TFC), and Total Anthocyanin Content (TAC)

The TPC was determined following the method described by Singleton, Orthofer, and Lamuela-Raventós method [12]. Initially, a gradient-diluted sample of 20 μL, 90 μL of water, and 10 μL of Folin–Ciocalteau reagent were added to a 96-well plate. The mixture was incubated in darkness at 25 °C for 5 min. Subsequently, 80 μL of a sodium carbonate solution (75 g/L) was added and thoroughly mixed. Finally, the absorbance was measured using a microplate reader (Epoch 2, BioTek Instruments, Inc., Winooski, VT, USA) at 765 nm. To quantify the TPC, a calibration standard curve was prepared, using gallic acid as the reference standard, and the TPC values were expressed as milligrams of gallic acid equivalent per gram of fresh weight (mg GAE/g FW).

The TAC was determined using the spectrophotometric pH differential method, as outlined by Giusti and Wrolstad [13]. Initially, a diluted sample was prepared with potassium chloride (0.025 M), and the pH of the solution was adjusted to 1.0 using hydrochloric acid. Subsequently, another diluted sample was prepared with sodium acetate (0.4 M), and the pH was adjusted to 4.5 using acetic acid. The absorbance was then measured at 510 nm and 700 nm using a microplate reader. TAC was calculated using Equation (4) and expressed as milligrams of cyanidin 3-glucoside (C3G) equivalent per 100 g of fresh weight (mg C3GE/100 g FW).
(4)$$\mathrm{TAC}=\frac{\left[{\left(A_{510}-A_{700}\right)}_{\mathrm{pH}1.0}-{\left(A_{510}-A_{700}\right)}_{\mathrm{pH}4.5}\right]\times \mathrm{MW}\times \mathrm{DF}\times V\times 10^{3}}{\varepsilon \times W\times 1}$$
where A_510_ and A_700_ refer to the absorption wavelengths at 510 nm and 700 nm, respectively; MW represents the molecular weight of C3G (449 g/mol); ε refers to the molar absorptivity of the reference anthocyanin (26,900); DF stands for the dilution factor; V refers to the extraction volume; and W represents the sample weight.

The TFC was measured using the aluminum chloride colorimetric assay, as described by Chang, Yang, Wen, and Chern [14], with slight modifications. In brief, a sample of 30 μL was incubated in a 96-well plate, along with 180 μL of water and 10 μL of NaNO_2_ (5%, *v*/*v*), for 6 min at 25 °C. Subsequently, 20 μL of AlCl_3_ (10%, *v*/*v*) was added and the mixture was shaken. Following another 6 min at 25 °C, 60 μL of 4% NaOH was added, and the solution was incubated at 37 °C for 15 min. The absorbance was monitored at 510 nm using a microplate reader. A standard curve was constructed using catechin, and the TFC was expressed as milligrams of catechin equivalent per gram of fresh weight (mg CE/g FW).

### 2.9. Determination of Ethylene (ETH) Content

The blue honeysuckle treated with either chitosan solution or aloe vera gel solution was evaluated for ETH measurement at 0, 3, 7, 14, 21, and 28 days. The samples were homogenized in a 0.1 M phosphate buffer solution (pH 7.4) and centrifuged at 3000× *g* for 20 min at 4 °C. The resulting supernatant was utilized to determine the ETH content using a Plant ETH ELISA Kit (Shanghai Yanqi Biotechnology Co., Ltd., Shanghai, China), in accordance with the manufacturer’s instructions.

### 2.10. Determination of Abscisic Acid (ABA) Content Using HPLC-DAD

The ABA content was determined using HPLC-DAD according to a previous method [15]. The sample was filtered through a 0.45 μm filter and then subjected to HPLC-DAD. A C18 column was employed for reverse-phase chromatography. The elution of ABA was carried out in an isocratic mode over a 20 min gradient with a mobile phase comprising 55% methanol, 40% 0.7% acetic acid, and 5% acetonitrile. The column temperature and the flow rate were set at 30 °C and 0.7 mL/min. Detection of ABA was achieved at a wavelength of 254 nm.

### 2.11. Determination of Antioxidant Capacity

Antioxidant capacity was determined by measuring the DPPH radical scavenging capacity, ABTS^•+^ radical scavenging capacity (ABTS), and ferric-reducing antioxidant power (FRAP). The DPPH radical scavenging capacity was determined using the method proposed by Brand-Williams, Cuvelier, and Berset [16]. In brief, 5 μL of the diluted sample was added to 195 μL of DPPH solution and thoroughly mixed. The mixture was then incubated in darkness at 25 °C for 2 h, and absorbance was recorded at 515 nm. The percentage of DPPH radical scavenging capacity was used to construct the Trolox calibration curve (100–1000 μmol/L), and the DPPH radical scavenging capacity was expressed as milligrams of Trolox equivalent per gram of fresh weight (mg TE/g FW). The ABTS^•+^ radical scavenging capacity was measured following the method reported by Re et al. [17]. Firstly, the ABTS solution was diluted with methanol to achieve an absorbance of 0.7 ± 0.01 at 734 nm, resulting in an ABTS^•+^ working solution (solution A). Then, 10 μL of the sample was added to 190 μL of solution A. Finally, the absorbance was monitored in a microplate reader at 734 nm. The ABTS^•+^ radical scavenging capacity percentage was used to plot the Trolox calibration curve (50–800 μmol/L) and expressed as milligrams of Trolox equivalent per gram of fresh weight (mg TE/g FW). The ferric-reducing antioxidant power was determined using the previous method [18]. In brief, the FRAP working solution (solution B) was freshly prepared by mixing 10 mM of TPTZ, 20 mM of FeCl_3_·6H_2_O, and 300 mM acetate buffer at a ratio of 1:1:10. Then, solution B (150 μL), sample (10 μL), and distilled water (30 μL) were added to a 96-well plate. The absorbance was monitored at 593 nm after incubating the reaction mixture for 30 min at 37 °C. The ferric-reducing antioxidant power percentage was used to plot the FeSO_4_·7H_2_O calibration curve and expressed as milligrams of FeSO_4_.7H_2_O equivalent per gram of fresh weight (mg FeSO_4_.7H_2_O/g FW).

### 2.12. Determination of Malondialdehyde (MDA) Content

The determination of MDA was carried out following the method reported by Li, Sun, Zhao, and Zhang [19]. Firstly, a sample of 30 μL was mixed with 10% trichloroacetic acid (450 μL), and the resulting mixture was then centrifuged at 1000× *g* for 10 min. Subsequently, 100 μL of the supernatant was added to a 0.6% sulfuric acid solution (100 μL). The absorbance was recorded at three wavelengths—450 nm, 532 nm, and 600 nm—using a microplate reader. MDA content was calculated according to Equation (5).
(5)$$Y\left(\mathrm{nmol}/g\right)=\frac{\left[6.45\times \left(A_{532}-A_{600}\right)-0.56\times A_{450}\;\right]\times V\times 1000}{w}$$
where A_532_, A_600_, and A_450_ represent the absorbance values at 532 nm, 600 nm, and 450 nm, respectively. Y refers to the MDA content; V is the sample volume; and W is the sample weight.

### 2.13. Statistical Analysis

The obtained results were subjected to ANOVA analysis, and significant differences at *p* ≤ 0.05 were determined using Duncan’s multiple range tests, performed using IBM SPSS Statistics 26.0 statistical data analysis software (SPSS Inc., Chicago, IL, USA). Principal component analysis, hierarchical cluster analysis, and Pearson’s correlation analysis were carried out using Origin 2021 v.9.8.0 200 software (Origin Lab Corporation, Northampton, MA, USA). All results are presented as mean ± standard deviation based on three independent experiments.

## 3. Results and Discussion

### 3.1. Effects of Different Treatments on Wax Coverage, Weight Loss, and Firmness

It is widely recognized that wax coverage, weight loss rate, and firmness are crucial parameters that reflect the freshness and texture of fruits. The transportation quality of blue honeysuckle is closely linked to these parameters. Wax plays a pivotal role and can impact reactive oxygen metabolism and membrane structure, as stated by Chu, Gao, Chen, Fang, and Zheng [20]. As shown in Table 1, the application of AL treatment effectively preserves the wax layer of blue honeysuckle. Although the wax coverage gradually decreased during 28 days of storage in both the control and coated treatments, the rate of decline in the AL treatment was comparatively slower. In stark contrast, the surface of the CH and CH+AL treatments exhibited little to no visible wax coverage due to the glycerol coating. Consequently, AL treatment emerges as an efficient method for maintaining the wax coverage of blue honeysuckle.

The weight of the fruit can significantly impact the commercial value of fresh blue honeysuckle. During the 28 days of storage, the weight loss percentage for all samples exhibited a steady increase during the initial 21 days, followed by a sharp escalation, culminating in its peak on day 28. By day 28, the weight loss in the control group had risen by 12.72%, while for blue honeysuckle coated with CH, AL, and CH+AL coatings, the weight loss had increased by 14.17%, 8.28%, and 1.11% compared to the values on day 21, respectively. Therefore, the impact of the AL coating treatment is noteworthy, with the most favorable outcomes being achieved through the combination of CH and AL. This phenomenon can be attributed to the formation of multi-layer coatings, which create a protective barrier, inhibiting moisture loss from blue honeysuckle from occurring through lenticels [21].

Weight loss represents the primary factor responsible for changes in firmness during the postharvest storage of berries. When weight loss reaches a certain threshold, firmness tends to increase sharply. As can be seen from Table 1, with a prolonged storage time, the firmness of blue honeysuckle exhibited a pattern of an initial increase, followed by a decrease, and then another increase. Firmness in the control group began to rise on day 21, while the treated samples started showing increased firmness on day 14. On day 28, compared to the initial values, blue honeysuckle in the control group and those coated with CH, AL, and CH+AL coatings had experienced firmness increases of 0.85 N, 0.48 N, 0.24 N, and 0.33 N, respectively. This may be due to the presence of a coating on the surface of blue honeysuckle acting as a physical barrier to oxygen exchange, thereby reducing metabolic activity. This, in turn, contributes to the delay in the softening process of the fruit and enhancements in its overall firmness [22].

### 3.2. Effects of Different Treatments on TSS, Acidity, and pH

TSS, acidity, and pH serve as essential indicators for evaluating the sensory quality of postharvest fruits. The various coating treatments have discernible effects on the TSS, acidity, and pH levels of blue honeysuckle during storage.

In our study, a modest and fluctuating increase in TSS content was observed over the 28-day storage period, with the highest TSS content being recorded on day 28 in the case of the CH+AL treatment. This rise in TSS content can be attributed to the reduction in moisture content and starch degradation that naturally occurs during storage [23]. The coating treatments effectively inhibited the respiration rate by creating a protective barrier around the fruit surface, sealing off stomata, inhibiting gas exchange, retarding the aging process, and protecting the TSS content [24].

Throughout the storage period, all samples displayed a declining trend in acidity, likely due to the utilization of organic acids in the respiratory metabolism [24]. In comparison to the acidity levels at day 0, it was observed that, after 28 days of storage, the control group exhibited a decrease of 47.3%, while the CH-, AL-, and CH+AL-coated treatments experienced reductions of 26.0%, 0.05%, and 29.6%, respectively. AL treatment demonstrated superior efficacy in preserving the acidity of blue honeysuckle, attributed to its film’s protective properties. This is consistent with the results for 0.5% AL-coated blueberries, which displayed slightly lower acidity than the control after storage [8].

Meanwhile, the pH of blue honeysuckle gradually increased in all treatments during the 28-day storage period. Vieira et al. found a gradual pH increase over a 25-day storage period in 0.5% chitosan- and 0.5% aloe vera gel-coated blueberries, which aligns with our results [8]. After 28 days, the pH in the control group reached a significantly higher level (pH 3.63) compared to the coated blue honeysuckle, indicative of lower stability. These results demonstrated the efficacy of coating in postponing the ripening and deterioration of blue honeysuckle, reaffirming the pH-protective effect of the coating [9].

### 3.3. Effects of Different Treatments on Respiration Rate

All fresh fruits respire after harvest; however, small fruits have non-climacteric behavior and reduced respiration after harvest and during the storage period. They absorb oxygen for metabolic processes and release carbon dioxide and water as metabolic by-products [25]. The primary objective of employing a coating is to decelerate the respiration rate during storage. During the 28-day storage period, the respiration rate of blue honeysuckle exhibited a pattern of an initial increase followed by a subsequent decrease, as illustrated in Table 1. The peak respiration rate was reached on day 14 for the control, CH, and AL treatments, while for the CH+AL treatment, it occurred on day 7. Notably, the coated samples consistently displayed lower respiration rates compared to the control, with the CH+AL treatment recorded the lowest rates among all samples after 28 days of storage. This outcome can be attributed to the formation of a film through the coating, which partially obstructed the lenticels and stomata on the fruit’s epidermis, thereby diminishing gas exchange [26].

### 3.4. Effects of Different Treatments on Color

Peel color represents a crucial physiological parameter in the post-harvest evaluation of blue honeysuckle. Figure 1 provides a clear visual insight into the changes in peel color during postharvest storage. Figure 1 also presents the alterations in the total color difference (Δ*E*) value of blue honeysuckle over the 28-day storage period. The Δ*E* value serves as a measure of the color difference, reflecting the changes in color between the two time points. Notably, a substantial increase in Δ*E* was observed in the control, while fruits coated with CH+AL exhibited a much smaller Δ*E* change (2.255 ± 0.04) during storage. This observation indicates that the multi-layer coating can effectively mitigate the color changes in blue honeysuckle. Li et al. have previously demonstrated that multi-layer coating treatments involving substances such as 1-methylcyclopropene and tea polyphenols are linked to fruit quality and can influence the peel color of bracken [6].

### 3.5. Effects of Different Treatments on Metabolite Profiling in Blue Honeysuckle

As widely recognized, fruits produce two distinct categories of metabolites: primary metabolites and secondary metabolites [27]. These metabolites in blue honeysuckle are pivotal indicators and serve as the foundation for evaluations of postharvest fruit quality. In our study, we focused on the primary metabolite, ASA, as well as various secondary metabolites, including TPC, TAC, TFC, ETH, and ABA.

ASA, as a primary metabolite, plays a crucial role as a non-enzymatic antioxidant in protecting plants from oxidative damage caused by abiotic stress [11]. Figure 2A,G illustrate the changes in ASA content during cold storage, revealing a gradual increase followed by a subsequent decrease. The initial fluctuating increase in ASA content can be attributed to the gradual ripening of the fruits after harvest, while the subsequent decrease may be due to the natural oxidation of ASA [9]. In terms of the final content, it is noteworthy that the CH+AL treatment exhibited the highest ASA content, reaching 5.85 mg/100 g FW, significantly higher than other treatments. This highlights the effectiveness of the multi-layer coating in preserving ASA levels in blue honeysuckle, as it forms a protective film on the surface that minimizes the infiltration of CO_2_ and O_2_, consequently reducing the autotrophic activity of ASA [26].

For the secondary metabolites including TPC, TAC, and TFC, Figure 2B presents the TPC of blue honeysuckle subjected to various treatments, revealing distinct patterns of change. In the control, CH, and AL treatments, TPC steadily increased from the onset of storage until the 7th day, followed by a gradual decline until the 28th day. In contrast, CH+AL-coated blue honeysuckle exhibited a gradual decrease in TPC. This observed increase in TPC can be attributed to the fact that blue honeysuckle cv. ‘Lanjingling’ is an early-maturing variety, harvested before reaching full ripeness to maximize the fruit’s shelf life in the fresh market. During storage, blue honeysuckle gradually produced phenolic compounds. Conversely, the decrease in TPC may result from fruit aging, the depletion of phenolic acids, and cell degradation [28]. Remarkably, during the 28-day storage period, the CH+AL treatment exhibited the highest TPC, measuring 17.87 mg GAE/g FW, significantly surpassing other treatments. This highlights the role of the chitosan coating, as demonstrated in previous studies, in reducing the loss of phenolic compounds and preventing fruit browning [9,28]. The AL coating may enhance the protective effect of TPC on the chitosan coating. In most cases, the TPC of the control was significantly higher than the treatment samples during the initial 14 days of storage. This can be attributed to the control fruit ripening faster than its treated counterparts at the beginning of the storage period. Blue honeysuckle is a berry known for its richness in anthocyanins, primarily C3G, a crucial parameter for evaluating postharvest quality [29]. Figure 2C illustrates the dynamics of TAC during the 28-day storage period in both the control and treatment samples. In all samples, TAC exhibited a gradual increase followed by a decline, with the peak occurring between days 3 and 7 of storage. This initial increase in anthocyanin levels may be linked to the reduction in fruit acidity, which provides a carbon framework for anthocyanin synthesis [30]. The subsequent decrease in TAC could be attributed to the presence of polyphenol oxidase, an enzyme catalyzing the oxidation of phenolic compounds into unstable quinones and forming polymers with anthocyanins [31]. In comparison to day 0, the TAC at day 28 decreased by 94.31% in the control, 72.72% in the CH treatment, 51.98% in the AL treatment, and 47.19% in the CH+AL-coated treatment. The coated treatments proved effective in preserving higher TAC levels in blue honeysuckle during prolonged cold storage, with the CH+AL-coated treatment exhibiting the most notable results. Ultimately, blue honeysuckle coated with CH+AL showed the highest TAC (562.15 mg C3GE/100 g) after 28 days of storage, significantly higher than other treatments and the control. This aligns with the findings of Nia et al., who demonstrated that postharvest AL-coated (25% and 33%) treatments had the potential to protect the TAC of table grapes [9]. TFC exhibited a consistent downward trend during the storage of blue honeysuckle, as illustrated in Figure 2D. The reduction in TFC after 28 days of storage had the following order: control > AL > CH+AL > CH, resulting in decreased values of 72.55%, 54.43%, 49.07%, and 48.55%, respectively, compared to the initial value at day 0. Notably, there was no statistically significant difference in the reduction between the CH- and CH+AL-coated treatments. These results indicated the effectiveness of coated treatments in preserving TFC in blue honeysuckle, particularly with the CH and CH+AL treatments. The application of a CH coating notably enhanced the preservation of TFC, a phenomenon supported by the findings of Jurić et al., who demonstrated that a 1% chitosan-based layer-by-layer edible coating was effective in protecting TFC and other bioactive compounds in mandarin fruit [32]. In summary, the CH+AL-coated treatment emerges as the optimal choice for preserving the postharvest TPC, TAC, and TFC of blue honeysuckle.

ETH is an important plant hormone, and its biosynthesis is strictly regulated during fruit ripening [33]. As shown in Figure 2E, the ETH content gradually increased during storage. The coated samples exhibited lower ETH levels compared to the control. This result is probably due to the formation of an O_2_ barrier on the surface of the coated blue honeysuckle, which inhibits ETH synthesis during the biosynthesis process [26]. The ETH content in the coated blue honeysuckle showed the following trends: AL > CH > CH+AL. The CH+AL-coated treatment has a pronounced preserving effect on blue honeysuckle and can inhibit ETH production. The results indicate that a layer-by-layer coating offers superior O_2_ barrier properties compared to single-layer application. Previous findings have demonstrated that a chitosan coating can reduce ETH production, and a layer-by-layer coating with 2% chitosan-montmorillonite nanocomposites can further suppress ETH production [5].

ABA is a crucial plant hormone, involved in fruit ripening and responses to adverse conditions [34]. As shown in Figure 2F,H, the ABA levels in blue honeysuckle gradually decreased during storage. Specifically, the ABA levels after 28 days significantly decreased in the following order: control > AL > CH+AL > CH, with reductions of 84.22%, 80.54%, 75.43%, and 64.87%, respectively, compared to the initial levels at day 0. Interestingly, the CH treatment performed the best, unlike ETH, as multiple coating layers may not have a cumulative effect on ABA content. While the coated treatment can reduce the rate of ABA decline in blue honeysuckle, these results may be attributed to the coated treatment delaying the accumulation of ABA and ABA conjugates. Bose, Howlader, Jia, Wang, and Yin discovered that the coated treatment using alginate oligosaccharides can suppress the expression of ABA signaling genes, delay ABA accumulation, and decrease ABA content in strawberries [35].

### 3.6. Effects of Different Treatments on Antioxidant Capacity and MDA Content

The antioxidant capacity of fruits is an important indicator to assess the nutritional quality of postharvest fruits [2]. As shown in Figure 3A, the DPPH free radical scavenging capacity was gradually increased during the 28-day storage of blue honeysuckle. The increase in the antioxidant potential of blue honeysuckle is attributed to changes in polyphenol composition during the ripening process [36]. The DPPH inhibition after 28 days for the control, CH, AL, and CH+AL increased by 57.26%, 91.51%, 47.81%, and 123.42%, respectively, compared to day 0. These findings suggest that CH-coated treatment can enhance DPPH antioxidant capacity, especially when combined with an AL coating. This enhancement might be due to the coating making it more challenging for blue honeysuckle to undergo oxidation during postharvest storage. Consistent with our results, Zhang, Zhao, Zhang, Sheng, Cao, and Jiang reported that a chitosan coating increased DPPH antioxidant capacity, with results exceeding 1.5 times that of the control during the storage of nectarine fruits [37]. Furthermore, they revealed that high-molecular-weight chitosan (120 kDa) exhibited a stronger DPPH antioxidant capacity than low-molecular-weight chitosan (30 kDa). Similarly, the trend in ABTS is consistent with that of DPPH during storage (Figure 3B). Over 28 days of storage, the final day ABTS values for the control, CH, AL, and CH+AL increased by 22.09%, 42.76%, 17.70%, and 26.35%, respectively, compared to day 0. The ABTS results indicated that CH had the best ability to scavenge free radical ABTS^·+^ in blue honeysuckle. Coatings with different molecular weights of chitosan also mitigated changes in antioxidants (mainly phenolic compounds), thereby maintaining the strong antioxidant capacity of postharvest nectarine fruit [37]. In accordance with de Souza et al. (2021), the use of natural edible films and coatings such as galactomannan, cashew gum, alginate, and gelatin helped to maintain the antioxidant capacity of grapes. Therefore, it is evident that the use of CH+AL as a fruit coating effectively increased the ABTS antioxidant capacity. As shown in Figure 3C, the FRAP-reducing power increased slightly during the first 7 days and then decreased during the 28-day storage period, differing from the results obtained with the ABTS and DPPH assays. These variations are attributed to the different mechanisms involved in radical antioxidant reactions [36]. High levels of FRAP-reducing power were observed in the control after 7 days of postharvest storage, which was twice as high as that in the treatments. This difference may be due to the higher levels of TPC, TAC, and TFC in the control after 7 days of storage. Previous studies have demonstrated a significant correlation between FRAP-reducing power and polyphenol content [5]. After 28 days of storage, the FRAP-reducing power of the control, CH, AL, and CH+AL decreased by 68.09%, 47.87%, 35.51%, and 43.95%, respectively, compared to day 0. These results indicate that coated treatments were more effective in preserving the FRAP-reducing power of blue honeysuckle, especially the AL coating. This effect is probably due to the significant antioxidant and antimicrobial potential of acemannan in AL [9]. In summary, this study demonstrates that coated treatments enhance the overall antioxidant capacity, including DPPH, ABTS, and FRAP, in postharvest blue honeysuckle fruit during storage. The antioxidant capacity of postharvest blue honeysuckle correlates with changes in coating methods (CH, AL, and CH+AL). The study shows that CH and CH+AL coatings significantly increase DPPH and ABTS antioxidant capacity, while the AL coating maintains the FRAP antioxidant capacity. These results may be attributed to coated treatments enhancing non-enzymatic antioxidative systems, thereby reducing reactive oxygen species (ROS) accumulation, improving fruit quality, and delaying senescence in postharvest fruits [38].

MDA serves as a biomarker for membrane damage resulting from lipid peroxidation in fruits during storage [38]. The MDA content in blue honeysuckle exhibited continuous accumulation over the 28-day storage period (Figure 3D). However, this increase in MDA in the fruit was significantly inhibited by CH, AL, and CH+AL coating treatments. Compared to the MDA levels at day 0, the values at day 28 for the control, CH, AL, and CH+AL increased by 88.71%, 72.70%, 83.00%, and 78.13%, respectively. Notably, when compared to AL, both CH and CH+AL coatings were more effective in reducing MDA content during the postharvest preservation of blue honeysuckle. These results indicate that coated treatments significantly mitigate lipid peroxidation in the cell membrane during the storage of blue honeysuckle, thereby enhancing its shelf life. Similarly, Zhang et al. found that a chitosan coating inhibited the accumulation of MDA in nectarines during storage compared to the control, thus improving lipid peroxidation in cell membranes during cold storage [37]. Consequently, it can be concluded that the CH-coated treatment was the most effective in reducing MDA accumulation.

### 3.7. Multivariate Statistical Analyses

In this study, to comprehensively evaluate the effectiveness of coated treatments on blue honeysuckle during cold storage, a principal component analysis, cluster analysis, and correlation analysis were applied. The relationship between samples and compositional variables was determined using these multivariate statistical analyses. The data collected for the samples were condensed into two principal components, which collectively explained 52.6% of the total variability. Specifically, PC1 accounted for 38.9%, and PC2 accounted for 13.7% of the variability (Figure 4A,B). Samples coated with CH, AL, and CH+AL were positioned in the second quadrant, indicating that the coating treatments were more effective in preserving blue honeysuckle compared to the control. Hierarchical cluster analysis further reinforced that the CH-coated treatment was the most effective, with CH-treated samples at 7 days being closely grouped with the control at day 0 (Figure 4C). To further evaluate the relationships among the measured parameters, a Pearson’s correlation analysis was conducted, with the results presented in Figure 4D,E. Firmness exhibited a positive correlation with weight loss, indicating that increased weight loss contributed to enhanced fruit firmness during storage. ABA demonstrated positive correlations with acidity (r = 0.82), TPC (r = 0.64), TAC (r = 0.83), TFC (r = 0.67), and FRAP (r = 0.76), while it showed a negative correlation with MDA (r = −0.55). These results demonstrate the significant role of ABA in shaping the fruit quality of postharvest blue honeysuckle. FRAP exhibited positive correlations with TPC (r = 0.92), TAC (r = 0.71), TFC (r = 0.69), and ABA (r = 0.76). DPPH displayed a strong positive correlation with ABTS (r = 0.90), but it exhibited negative correlations with TPC (r = −0.36), TAC (r = −0.45), and TFC (r = −0.44). This suggests that antioxidant capacity is closely linked to metabolites. Additionally, MDA displayed a negative correlation with FRAP. These results collectively highlight that the senescence of blue honeysuckle is accompanied by an increase in ROS levels and a decrease in overall quality. This is in line with the findings of Li et al., who reported a similar correlation between ROS metabolism and fruit senescence in postharvest bracken [6].

To the best of our knowledge, this study represents the first research on coated treatments for blue honeysuckle during storage. The results of our correlation analysis confirmed the close connections between antioxidant capacity and metabolites. Numerous studies have indicated that TPC can enhance antioxidant capacity [1], thereby inhibiting the production of free radicals and activating the intracellular antioxidant defense system [6]. Additionally, coated treatments can create a microenvironment on the surface of blue honeysuckle, which can slow down the respiration rate and influence the physiological metabolism of postharvest blue honeysuckle. Therefore, they reduce the accumulation of intracellular oxidative stress products (MDA) and ultimately mitigate oxidative stress damage (Figure 5) [26].

## 4. Conclusions

This study evaluated the effective role of postharvest applications of CH and AL in blue honeysuckle. Based on our results, coating blue honeysuckle with CH and AL leads to increased firmness, TSS, acidity, pH, and antioxidant capacity (measured by DPPH, ABTS, and FRAP), thus enhancing fruit quality.

Additionally, these coatings helped maintain the levels of secondary metabolites (TPC, TAC, and TFC) while reducing weight loss, respiration rate, total color difference, and ABA, ETH, and MDA contents. Furthermore, the CH+AL coating proved to be the most effective treatment, as indicated by the values of ASA, TPC, TAC, TFC, ABA, and antioxidant capacity (measured by DPPH and ABTS). In summary, considering all the measured parameters influenced by different coating materials, the layer-by-layer (CH followed by AL) coating approach holds promise as a valuable technique to maximize the beneficial effects of blue honeysuckle when used as an edible coating.

## Figures and Tables

**Figure 1 foods-13-00630-f001:**
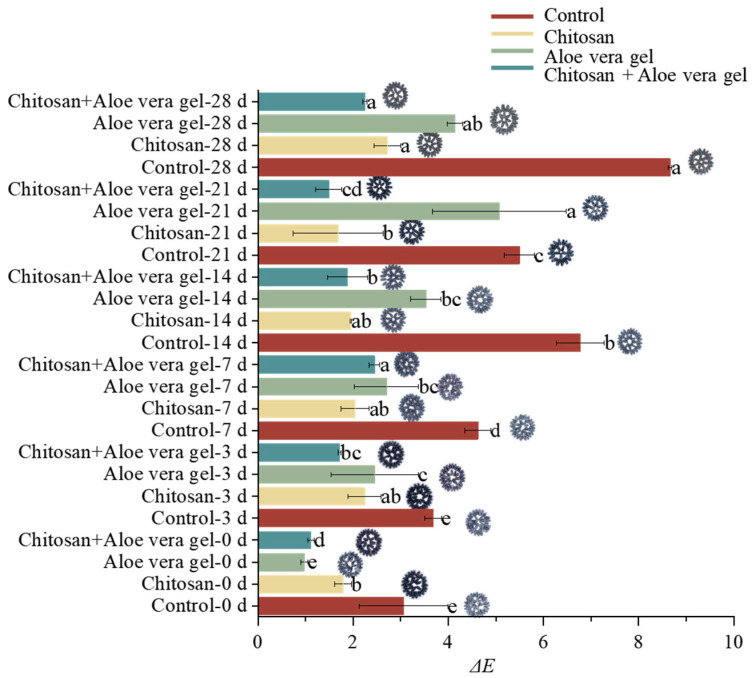
Comparison of the color difference (Δ*E*) in blue honeysuckle treated with chitosan (CH), aloe vera gel (AL), and chitosan + aloe vera gel (CH+AL) versus the control during 28 days of storage at −1 °C. The lowercase letters indicate significant differences among storage days for each parameter, based on ANOVA test. The different letters (a–e) indicate significant differences between treatments using Duncan’s multiple range test.

**Figure 2 foods-13-00630-f002:**
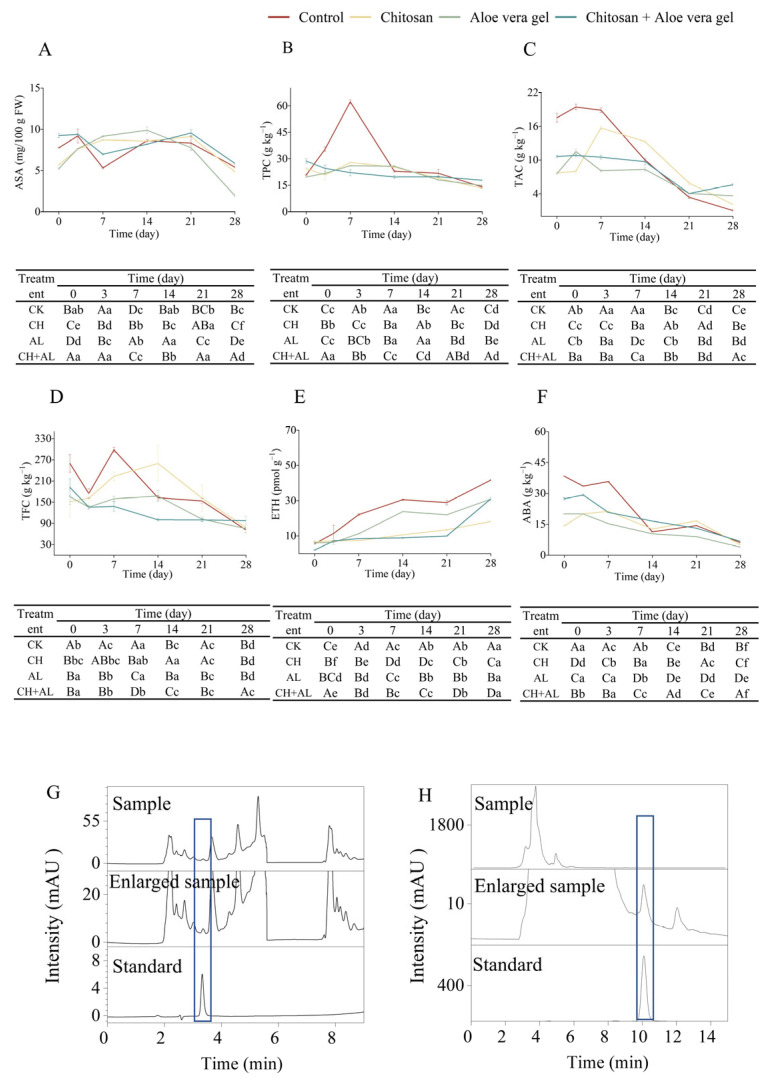
Comparison of ASA (**A**), TPC (**B**), TAC (**C**), TFC (**D**), ETH (**E**), ABA (**F**), ASA chromatogram (Blue rectangle: time-to-Peak force of ASA) (**G**), and ABA chromatogram (Blue rectangle: time-to-Peak force of ABA) (**H**) in blue honeysuckle treated with chitosan (CH), aloe vera gel (AL), and chitosan + aloe vera gel (CH+AL) versus the control during 28 days of storage at −1 °C. The upper case letters indicate significant differences among four treatment, and the lowercase letters indicate significant differences among storage days for each parameter, based on ANOVA test.

**Figure 3 foods-13-00630-f003:**
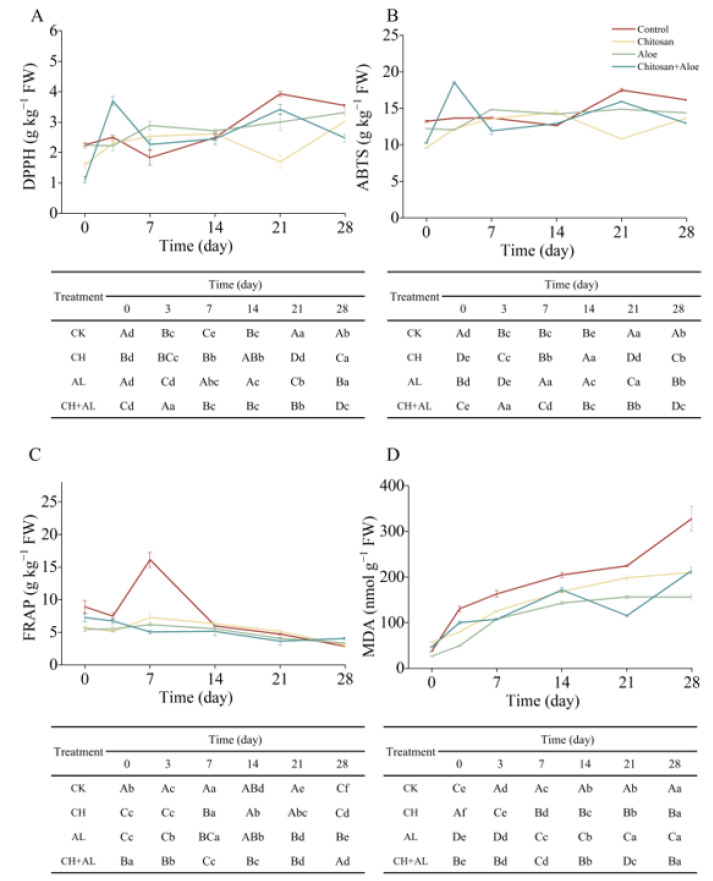
Assessment of DPPH (**A**), ABTS (**B**), FRAP (**C**), antioxidant capacity, and MDA content (**D**) in blue honeysuckle treated with chitosan (CH), aloe vera gel (AL), and chitosan + aloe vera gel (CH+AL) versus the control during 28 days of storage at −1 °C. The upper case letters indicate significant differences among four treatment, and the lowercase letters indicate significant differences among storage days for each parameter, based on ANOVA test.

**Figure 4 foods-13-00630-f004:**
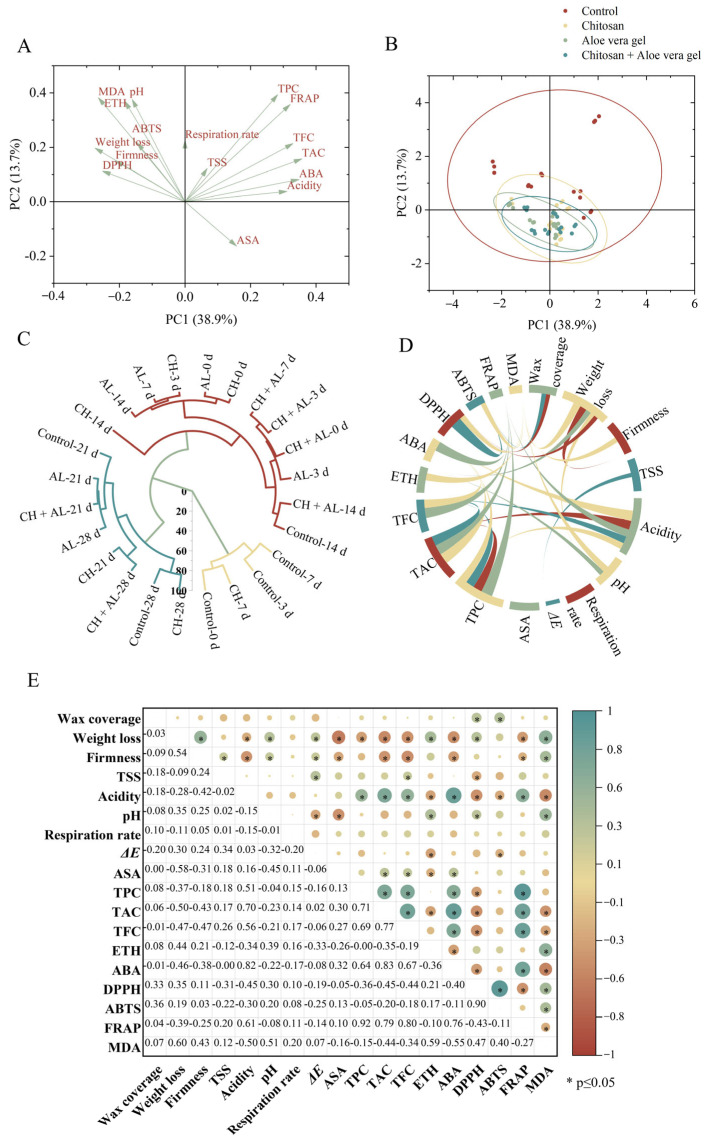
Multivariate data analyses, including PAC scores (**A**,**B**), hierarchical cluster analysis (**C**), chord diagram (**D**), and correlation analysis (**E**). Green indicates a positive correlation, while brown indicates a negative correlation. Deeper colors indicate stronger correlations. Asterisks (*) indicates significant correlation at *p* ≤ 0.05.

**Figure 5 foods-13-00630-f005:**
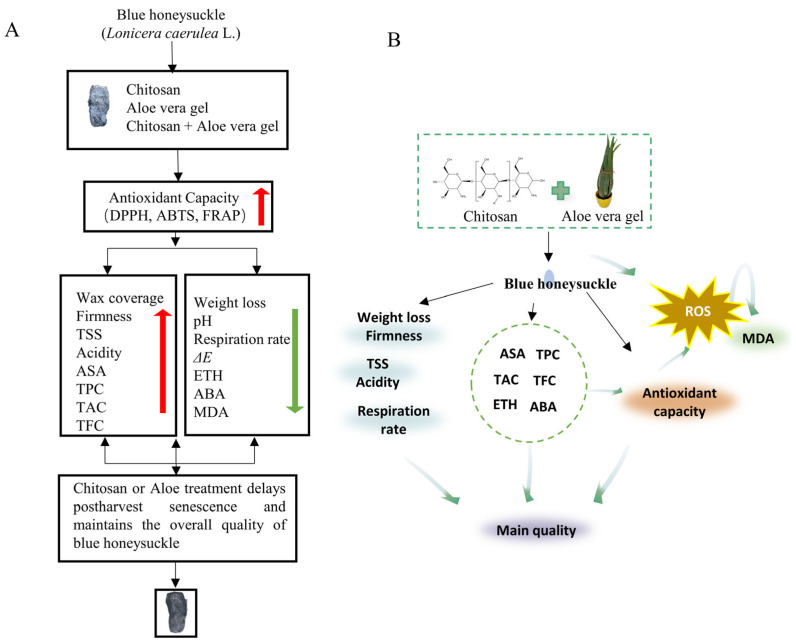
The mechanism diagram of coated blue honeysuckle, text version (**A**), and drawing version (**B**) illustration of the potential mechanism behind the edible coating treatments using CH and AL solution, which delay postharvest senescence and maintain the overall quality of blue honeysuckle by regulating the ROS metabolism. Red arrows represent promotional effects, while green arrows represent inhibitory effects.

**Table 1 foods-13-00630-t001:** Wax coverage, weight loss, firmness, TSS, acidity, pH, and respiration rate of blue honeysuckle during cold storage.

	Day	Control	Chitosan	Aloe Vera Gel	Chitosan + Aloe Vera Gel
Wax coverage (%)	0	80.42% ± 0.001 ^Aa^	0.00 ± 0.000 ^Ca^	57.08% ± 0.003 ^Ba^	0.00 ± 0.000 ^Ca^
	3	55.83% ± 0.003 ^Ab^	0.00 ± 0.000 ^Ca^	54.17% ± 0.002 ^Bb^	0.00 ± 0.000 ^Ca^
	7	47.92% ± 0.002 ^Bc^	0.00 ± 0.000 ^Ca^	52.92% ± 0.003 ^Ac^	0.00 ± 0.000 ^Ca^
	14	44.17% ± 0.002 ^Bd^	0.00 ± 0.000 ^Ca^	51.67% ± 0.004 ^Ad^	0.00 ± 0.000 ^Ca^
	21	34.58% ± 0.001 ^Bf^	0.00 ± 0.000 ^Ca^	48.33% ± 0.004 ^Ae^	0.00 ± 0.000 ^Ca^
	28	37.08% ± 0.001 ^Be^	0.00 ± 0.000 ^Ca^	46.25% ± 0.001 ^Af^	0.00 ± 0.000 ^Ca^
Weight loss (%)	3	−0.02% ± 0.001 ^Bb^	0.25% ± 0.001 ^ABc^	−3.34% ± 0.002 ^Cb^	0.49% ± 0.002 ^Ac^
	7	0.22% ± 0.002 ^Ab^	0.28% ± 0.003 ^Ac^	−2.74% ± 0.027 ^Bc^	0.58% ± 0.006 ^Ac^
	14	0.36% ± 0.002 ^Cb^	0.81% ± 0.001 ^Bb^	−2.33% ± 0.003 ^Dd^	1.15% ± 0.001 ^Ab^
	21	0.45% ± 0.002 ^Bb^	0.84% ± 0.003 ^ABb^	−1.17% ± 0.002 ^Ce^	1.22% ± 0.002 ^Ab^
	28	13.17% ± 0.005 ^Ba^	15.01% ± 0.003 ^Aa^	7.11% ± 0.003 ^Ca^	2.33% ± 0.003 ^Da^
Firmness (N)	0	0.63 ± 0.04 ^Cd^	1.05 ± 0.06 ^ABc^	0.95 ± 0.08 ^Bcd^	1.10 ± 0.03 ^Ac^
	3	0.88 ± 0.03 ^Bc^	0.91 ± 0.02 ^Bd^	1.37 ± 0.08 ^Aa^	1.33 ± 0.03 ^Aab^
	7	1.19 ± 0.05 ^Ab^	0.97 ± 0.03 ^Bcd^	0.93 ± 0.03 ^Bcd^	0.96 ± 0.06 ^Bd^
	14	1.28 ± 0.10 ^Ab^	0.87 ± 0.04 ^Bd^	0.79 ± 0.07 ^Bd^	0.95 ± 0.11 ^Bd^
	21	0.83 ± 0.03 ^Dc^	1.36 ± 0.06 ^Ab^	1.02 ± 0.01 ^Cbc^	1.22 ± 0.02 ^Bb^
	28	1.48 ± 0.05 ^Aa^	1.53 ± 0.10 ^Aa^	1.19 ± 0.23 ^Bab^	1.43 ± 0.08 ^ABa^
TSS (Brix°)	0	11.87 ± 0.06 ^Cd^	14.50 ± 0.40 ^Aa^	11.50 ± 0.10 ^Cb^	13.13 ± 0.50 ^Bbc^
	3	12.20 ± 0.44 ^Bcd^	10.80 ± 0.10 ^Cc^	13.47 ± 0.38 ^Aa^	13.30 ± 0.10 ^Aab^
	7	13.43 ± 0.42 ^Aa^	12.63 ± 0.15 ^Bb^	11.37 ± 0.31 ^Cb^	11.77 ± 0.15 ^Cd^
	14	13.43 ± 0.31 ^BCa^	14.67 ± 0.23 ^Aa^	13.70 ± 0.20 ^Ba^	12.80 ± 0.70 ^Cbc^
	21	12.73 ± 0.12 ^Bb^	14.33 ± 0.25 ^Aa^	11.30 ± 0.17 ^Db^	12.27 ± 0.29 ^Ccd^
	28	12.53 ± 0.06 ^Bbc^	13.13 ± 0.61 ^Bb^	11.37 ± 0.15 ^Cb^	14.13 ± 0.74 ^Aa^
Acidity (%)	0	1.46 ± 0.08 ^Aa^	1.17 ± 0.03 ^Ba^	1.04 ± 0.03 ^Ca^	1.15 ± 0.05 ^Bb^
	3	1.40 ± 0.07 ^Aab^	1.17 ± 0.03 ^Ba^	0.95 ± 0.03 ^Cb^	1.11 ± 0.10 ^Bb^
	7	1.33 ± 0.06 ^Ab^	1.04 ± 0.07 ^Bb^	0.94 ± 0.04 ^Cb^	1.32 ± 0.03 ^Aa^
	14	0.96 ± 0.05 ^Ac^	0.91 ± 0.04 ^Ac^	1.00 ± 0.01 ^Aab^	0.95 ± 0.08 ^Ac^
	21	0.96 ± 0.05 ^ABc^	1.02 ± 0.02 ^Ab^	0.96 ± 0.04 ^ABb^	0.88 ± 0.05 ^Bcd^
	28	0.77 ± 0.02 ^Bd^	1.06 ± 0.04 ^Ab^	0.77 ± 0.04 ^Bc^	0.81 ± 0.03 ^Bd^
pH	0	3.41 ± 0.01 ^Ac^	3.38 ± 0.01 ^Bb^	3.37 ± 0.01 ^Bc^	3.22 ± 0.01 ^Cd^
	3	3.44 ± 0.02 ^Ac^	3.25 ± 0.01 ^Cd^	3.38 ± 0.02 ^Bbc^	3.24 ± 0.01 ^Dd^
	7	3.44 ± 0.01 ^Ac^	3.22 ± 0.01 ^De^	3.27 ± 0.01 ^Cd^	3.30 ± 0.01 ^Bc^
	14	3.47 ± 0.01 ^Ab^	3.29 ± 0.02 ^Cc^	3.26 ± 0.01 ^Dd^	3.30 ± 0.01 ^Bc^
	21	3.53 ± 0.00 ^Aa^	3.31 ± 0.01 ^Dc^	3.40 ± 0.01 ^Bab^	3.37 ± 0.02 ^Cb^
	28	3.63 ± 0.01 ^Aa^	3.40 ± 0.01 ^Ca^	3.41 ± 0.01 ^Ba^	3.49 ± 0.02 ^Ba^
Respiration rate (mg CO_2_/kg • h)	0	39.56 ± 0.35 ^Bd^	43.00 ± 3.63 ^Bc^	54.79 ± 0.72 ^Ac^	24.59 ± 1.26 ^Cd^
	3	47.29 ± 1.30 ^Cc^	69.34 ± 3.92 ^Aa^	72.67 ± 2.36 ^Ab^	54.75 ± 4.47 ^Bb^
	7	70.80 ± 4.08 ^Ab^	74.70 ± 1.72 ^Aa^	73.22 ± 1.82 ^Ab^	66.04 ± 1.38 ^Ba^
	14	120.13 ± 3.78 ^Aa^	74.86 ± 6.55 ^Ba^	80.58 ± 4.96 ^Ba^	42.57 ± 1.90 ^Cc^
	21	42.88 ± 2.10 ^Ccd^	45.26 ± 0.59 ^BCc^	52.94 ± 1.34 ^Acd^	46.73 ± 1.39 ^Bc^
	28	65.80 ± 6.18 ^Ab^	53.69 ± 1.90 ^Bb^	49.67 ± 0.93 ^Bd^	42.77 ± 1.36^Cc^

The data are presented as mean ± standard deviation. Duncan’s multiple range test at (*p* ≤ 0.05) was employed to assess the mean separation for significant analysis of variance within both the columns (a, b, c, etc.) and rows (A, B, C, etc.).

## Data Availability

The original contributions presented in the study are included in the article, further inquiries can be directed to the corresponding author.

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
