# Peer review of "Enhancing Postharvest Quality and Antioxidant Capacity of Blue Honeysuckle cv. ‘Lanjingling’ with Chitosan and Aloe vera Gel Edible Coatings during Storage"

_foods, 2024, doi:10.3390/foods13040630_

Round 1
Reviewer 1 Report
Comments and Suggestions for Authors
Dear author,
Thanks for your work and report. My comments and questions are included in the attached file.
Regards

Comments on the Quality of English Language
Not too bad.
Author Response
[Response to Reviewers’ comments]
Reviewer #1: Comments and Suggestions for Authors
Dear author,
Thanks for your work and report. My comments and questions are included in the attached file.
Comments on the Quality of English Language
Not too bad.
[Response] Thank you so much for your support. We have responded to all the suggestions and questions you raised in the PDF file.

Reviewer 2 Report
Comments and Suggestions for Authors
Dear Authors,
research that opens new paths for the development of effective edible coatings for post-harvest preservation of fruits with their positive impact on their quality parameters is very important. The presented research work with its results offers valuable knowledge for the selection of suitable coatings for blue honeysuckle. Very valuable are the results of monitoring changes in the content of primary and secondary metabolites during storage, as well as changes in antioxidant activity using three analytical methods, which are the most widely used in current analyses.
The work is processed at a high level, the results are relevant and contain all important data.
I only make two additional comments:
1. correct the names "total phenols" in the text to "total polyphenols",
2. you calculated the results of the contents of the monitored parameters for the fresh mass. It is known that the dry matter of plant material changes during storage, so please add to the methodology whether you also took into account changes in the dry matter of honeysuckle during the calculation.
After supplementing my comments, I propose to believe the scientific work.
Author Response
Reviewer #2: Comments and Suggestions for Authors
Dear authors, research that opens new paths for the development of effective edible coatings for post-harvest preservation of fruits with their positive impact on their quality parameters is very important. The presented research work with its results offers valuable knowledge for the selection of suitable coatings for blue honeysuckle. Very valuable are the results of monitoring changes in the content of primary and secondary metabolites during storage, as well as changes in antioxidant activity using three analytical methods, which are the most widely used in current analyses.
The work is processed at a high level, the results are relevant and contain all important data.
[Response] Thank you so much for your support.
I only make two additional comments:
- correct the names "total phenols" in the text to "total polyphenols",
[Response] Thank you for your suggestion. We have searched for "total phenols" in this manuscript, and the results indicated that "total phenols" are referenced. Therefore, we have modified "total phenols" to "total polyphenol content" in line 176, and they are highlighted in yellow.
- you calculated the results of the contents of the monitored parameters for the fresh mass. It is known that the dry matter of plant material changes during storage, so please add to the methodology whether you also took into account changes in the dry matter of honeysuckle during the calculation.
[Response] Thank you for your suggestion. It is acknowledged that the dry matter content of plant material undergoes changes during storage. We examined the changes in dry matter content of blue honeysuckle during storage. The dry matter content of blue honeysuckle gradually decreases during storage, with a reduction of less than 4% over a period of 28 days. However, in analyses involving fresh weight, dry weight is typically not factored into the calculations and is instead considered as an additional indicator. Consequently, dry weight was not chosen for analysis in this experiment.
After supplementing my comments, I propose to believe the scientific work.
[Response] Thank you so much for your support.

Reviewer 3 Report
Comments and Suggestions for Authors
It is a well designed, performed and written piece of work.
There is but a few points for further clarification.
1. which are the 'optimal commercial ripeness' (page 99) harvest maturity indices.
2. All figures, statistics ( mean separation according to Duncan's multiple range test) are comparing the means of each treatment along storage time (or DE for Fig. 1). As consequence in results, details (such peaks e.g. p. 380) which may not have a significance are analysed and discussed, while the comparison between treatments where no statistics are available the trends of parameters during storage are follow mostrly the same pattern. Therefore, statistics to compare treatments either within storage time or a general LSD for the whole figure of each parameter is neccesary.
Individual treatment figures should be omitted and provide only the one with all treatments (in the bottom) with the appropriate statistics.
Similaly, the bar graph should be transformed to line graph to facilitate reading
Author Response
Reviewer #3: Comments and Suggestions for Authors
It is a well designed, performed and written piece of work.
[Response] Thank you so much for your support.
There is but a few points for further clarification.
- which are the 'optimal commercial ripeness' (page 99) harvest maturity indices.
[Response] Thank you. The term 'optimal commercial ripeness' for blue honeysuckle refers to 45 days after fruit flowering. We have included the specific harvest time for blue honeysuckle. These additions are highlighted in yellow; please refer to line numbers 102 for clarification.
- All figures, statistics (mean separation according to Duncan's multiple range test) are comparing the means of each treatment along storage time (or DE for Fig. 1). As consequence in results, details (such peaks e.g. p. 380) which may not have a significance are analysed and discussed, while the comparison between treatments where no statistics are available the trends of parameters during storage are follow mostrly the same pattern. Therefore, statistics to compare treatments either within storage time or a general LSD for the whole figure of each parameter is neccesary.
[Response] Thank you for your suggestion. We have incorporated statistical analysis between treatments in Figures 2 and Figure 3, and elaborated on this aspect in the article. These additions are highlighted in yellow; please refer to line numbers 382, 386, 401, 423, and 512 for further details.
Individual treatment figures should be omitted and provide only the one with all treatments (in the bottom) with the appropriate statistics.
[Response] Thank you for your suggestion. We have removed the individual treatment figures and included only the one featuring all treatments. These changes are highlighted in yellow; please refer to line number 386 for further details.
Similaly, the bar graph should be transformed to line graph to facilitate reading.
[Response] Thank you for your suggestion. We have converted the bar graph to a line graph. These modifications are highlighted in yellow; please refer to line numbers 512 for clarification.

Round 2
Reviewer 1 Report
Comments and Suggestions for Authors
Dear author,
Thanks for your revised file.
Regards
Comments on the Quality of English Language
It is OK.